# Shape Sensing and Kinematic Control of a Cable-Driven Continuum Robot Based on Stretchable Capacitive Sensors

**DOI:** 10.3390/s24113385

**Published:** 2024-05-24

**Authors:** Wenjun Shen, Jianhui He, Guilin Yang, Xiangjie Kong, Haotian Bai, Zaojun Fang

**Affiliations:** 1Zhejiang Key Laboratory of Robotics and Intelligent Manufacturing Equipment Technology, Ningbo Institute of Materials Technology and Engineering, Chinese Academy of Sciences, Ningbo 315201, China; shenwenjun@nimte.ac.cn (W.S.); hejianhui@nimte.ac.cn (J.H.); kongxiangjie@nimte.ac.cn (X.K.); baihaotian@nimte.ac.cn (H.B.); fangzaojun@nimte.ac.cn (Z.F.); 2College of Materials Science and Opto-Electronic Technology, University of Chinese Academy of Sciences, Beijing 100049, China; 3Nottingham Ningbo China Beacons of Excellence Research and Innovation Institute, School of Computer Science, University of Nottingham, Ningbo 315199, China

**Keywords:** cable-driven continuum robot, stretchable capacitive sensors, shape-sensing model, closed-loop control

## Abstract

A Cable-Driven Continuum Robot (CDCR) that consists of a set of identical Cable-Driven Continuum Joint Modules (CDCJMs) is proposed in this paper. The CDCJMs merely produce 2-DOF bending motions by controlling driving cable lengths. In each CDCJM, a pattern-based flexible backbone is employed as a passive compliant joint to generate 2-DOF bending deflections, which can be characterized by two joint variables, i.e., the bending direction angle and the bending angle. However, as the bending deflection is determined by not only the lengths of the driving cables but also the gravity and payload, it will be inaccurate to compute the two joint variables with its kinematic model. In this work, two stretchable capacitive sensors are employed to measure the bending shape of the flexible backbone so as to accurately determine the two joint variables. Compared with FBG-based and vision-based shape-sensing methods, the proposed method with stretchable capacitive sensors has the advantages of high sensitivity to the bending deflection of the backbone, ease of implementation, and cost effectiveness. The initial location of a stretchable sensor is generally defined by its two endpoint positions on the surface of the backbone without bending. A generic shape-sensing model, i.e., the relationship between the sensor reading and the two joint variables, is formulated based on the 2-DOF bending deflection of the backbone. To further improve the accuracy of the shape-sensing model, a calibration method is proposed to compensate for the location errors of stretchable sensors. Based on the calibrated shape-sensing model, a sliding-mode-based closed-loop control method is implemented for the CDCR. In order to verify the effectiveness of the proposed closed-loop control method, the trajectory tracking accuracy experiments of the CDCR are conducted based on a circle trajectory, in which the radius of the circle is 55mm. The average tracking errors of the CDCR measured by the Qualisys motion capture system under the open-loop and the closed-loop control are 49.23 and 8.40mm, respectively, which is reduced by 82.94%.

## 1. Introduction

A Cable-Driven Continuum Robot (CDCR) is a multi-degree-of-freedom mechanism actuated by light cables, which consists of a number of identical Cable-Driven Continuum Joint Modules (CDCJMs) [1,2,3]. Each CDCJM is composed of a base platform, a pattern-based flexible backbone, four driving cables, and a moving platform. Supported by the flexible backbone, the 2-DOF bending motions of the CDCJM are realized by controlling the driving cable lengths. The CDCR has the advantages of high flexibility and good adaptability. Therefore, it has been widely applied for dexterous manipulation in confined and complex spaces, such as minimally invasive surgery [4,5,6,7], collapsed buildings [8], space station mock-up environments [9,10], and the on-wing inspection of gas turbine engines [11,12].

A CDCR generally adopts a flexible backbone as its passive compliant joint to realize the 2-DOF bending deflections. Due to the disadvantage of low stiffness, the bending deflection of the flexible backbone is determined by not only the driving cable lengths but also the gravity and payload, which results in an inaccurate kinematic model of the CDCR. Therefore, the shape sensing of the flexible backbone is significant for a CDCR, since employs external measurement devices to measure the bending shape of the flexible backbone.

The commonly used shape-sensing methods of the continuum robots include the electromagnetic (EM)-tracking-based shape-sensing method [13,14,15], the vision-based shape-sensing method [16,17,18], and the shape-sensing method based on Fiber Bragg Grating (FBG) sensor [19,20,21]. The EM-tracking-based method employs an EM tracking system to simultaneously realize tip tracking and the shape measurement of the continuum robots. The EM tracking system consists of the EM field generator and multiple distributed EM sensors, in which the EM sensors are located along the continuum robots. Based on the position and orientation of each EM sensor, shape reconstruction algorithms are proposed in conjunction with the kinematic models of continuum robots. In [22], an extended Kalman filter is employed for the shape estimation of a surgical snake robot based on the EM sensor data. In [23], the shape reconstruction algorithm based on the quadratic Bezier curves is conducted for a CDCR, in which the pose information measured through the EM sensors and the length information of the robot are utilized to fit the shape of the robot. The shape-sensing method based on EM sensors is easy to integrate due to the small size of each EM sensor. However, it will result in the distortion of the EM field when there exist magnetic and conductive objects, which will decrease the measurement accuracy of the EM tracking system. Furthermore, the EM tracking method has limited workspace, and its tracking accuracy varies with the distance from the center of the EM field.

The vision-based shape-sensing method employs stereo cameras, infrared cameras, or high-speed cameras to measure the shape of the CDCR. In [24], the shape-sensing method based on stereo vision is proposed, which employs the Self-Organizing Mapping (SOM) algorithm for the shape reconstruction of a continuum robot based on the point cloud derived by the cameras. In [25], a marker-based shape-sensing method is proposed, in which multiple markers are located along the continuum robot. The positions of markers are measured through cameras, which are utilized to calculate the actual shape of the continuum robot. Although the vision-based shape-sensing method has high measurement accuracy, the measurement results are significantly affected by the external environment, which limits its application in confined spaces. Furthermore, the vision-based shape-sensing method has a serious time delay, which is not conducive to real-time control of a CDCR.

The FBG-sensor-based shape-sensing method employs the changes in fiber wavelengths to estimate the shape of continuum robots. In [21], the FBG fiber is inserted into the continuum robot with 1-DOF bending motion, in which the strain of the FBF fiber is employed to calculate the curvature of the continuum robot. In [26], the helical FBG sensors are located on the continuum robot, which are utilized for the curvature, torsion, and force measurement. The FBG-sensor-based shape-sensing method has the advantages of high resolution, high sensitivity, and high signal-to-noise ratio. However, the FBG sensors have to employ the interrogation system to demodulate the wavelengths of fibers, which is not conducive to integration.

The shape-sensing methods mentioned above cannot simultaneously have the advantages of high measuring accuracy, strong anti-interference capability, and the ability to measure the large bending deflection of the flexible backbone in confined spaces. Therefore, the shape-sensing method based on stretchable capacitive sensors is employed in this work. The stretchable capacitive sensors are a class of elastic strain sensors, which are made of flexible fabric and conductive nanomaterial to realize strain detection. They have the advantages of a high stretch rate, high sensitivity, and high stability. The capacitive value of the sensor will increase with its stretched length. The conventional shape-sensing methods based on stretchable sensors employ the parallel sensor location scheme, i.e., the location directions of sensors are parallel to the axis direction of the flexible backbone [27]. Such a sensor location scheme usually adopts four stretchable sensors for the shape measurement of the backbone with the 2-DOF bending deflections, which has good robustness. When the backbone produces the bending deflections, the shapes of sensors are assumed as a circle attached to the surface of the backbone, which can simplify the shape-sensing modeling analysis. However, the accuracy of the shape-sensing model with such a sensor location scheme will significantly decrease when there are location position errors of stretchable sensors. Furthermore, the parallel location scheme will increase the number of sensors.

In this paper, shape-sensing and closed-loop control based on stretchable capacitive sensors is proposed for the CDCR with a pattern-based flexible backbone. For the CDCJM with 2-DOF bending motions, two stretchable capacitive sensors are employed to measure the bending deflection of the flexible backbone so as to calculate the joint variables of the CDCJM. Two endpoints of each sensor are located on the surface of the backbone, and the location of each sensor can be described by four parameters. A generic shape-sensing modeling method is developed based on the 2-DOF bending deflections of the backbone, i.e., the relationship between the stretched lengths of two sensors and two joint variables. Due to the location errors of sensors, the calibration is conducted to further increase the accuracy of the shape-sensing model and realize the accurate shape measurement of the CDCR. The proposed shape-sensing method is insensitive to the location position errors of sensors, which can develop an accurate shape-sensing model with fewer sensors compared with the shape-sensing method applying the parallel sensor location scheme. Therefore, it has the advantages of high accuracy, high resolution ratio, and low cost. Based on the calibrated shape-sensing model, the sliding-mode-based closed-loop control is implemented for the CDCR.

The rest of this paper is organized as follows. Section 2 presents the configuration design of the CDCR with a pattern-based flexible backbone. Section 3 addresses the kinematic analysis of the CDCJM and the CDCR, including the displacement analysis, the velocity analysis, and the inverse kinematic analysis. Section 4 presents the shape-sensing modeling method of the CDCJM based on two stretchable capacitive sensors and the calibration method to compensate for the location errors of sensors. Furthermore, the closed-loop control is conducted for the CDCR based on the calibrated shape-sensing model. In Section 5, experiments on the CDCJM and the CDCR are conducted to validate the effectiveness of the proposed shape-sensing model and the closed-loop control method. The conclusion of this paper is given in Section 6.

## 2. Configuration Design of the Cable-Driven Continuum Robot

As shown in Figure 1, a CDCR usually employs a modular design method, which is composed of a set of identical serially connected CDCJMs. Each CDCJM merely allows 2-DOF bending motions, which consist of a base platform, a moving platform, a flexible backbone, and driving cables referring to Figure 2. The moving platform of the $i^{\mathrm{th}}$ CDCJM is the base platform of the ${\left(i+1\right)}^{\mathrm{th}}$ CDCJM. The driving cables are evenly mounted on the base and moving platforms, respectively. The flexible backbone is fixed at the centers of the base and moving platforms.

In this paper, a pattern-based flexible backbone is employed for the CDCJM, which possesses low bending stiffness but high tensile stiffness and high torsion stiffness to achieve the designated 2-DOF bending motions of the CDCJM. The rectangular patterns employed are inspired by elastic couplings. As shown in Figure 3, each patterned segment has two rectangular patterns. The structure parameters of the pattern-based flexible backbone include the inner diameter $d_{1}$, the thickness *t*, the width of the pattern *a*, the distance between two adjacent patterns *l*, and the central angle subtended by the pattern β. Since the existing stiffness modeling methods of the patterned-based flexible backbone cannot develop accurate analytical stiffness models when the flexible backbone produces large bending deflections, an FEA-based data-driven parameter stiffness modeling method is proposed. Such a stiffness modeling method uses a set of structure parameters within their dimension bounds and the simulation stiffness values computed by the FEA software to train the stiffness model. The Gaussian Process Regression (GPR) is employed due to its capability of solving nonlinear regression problems with fewer training data. Based on the trained stiffness models, the structure parameter optimization of the flexible backbone is conducted using the particle swarm optimization method to minimize the ratio of bending stiffness to tensile stiffness and the ratio of bending stiffness to torsion stiffness. The optimized structure parameters are given as follows: $d_{1}=9\;\mathrm{mm}$, t=4mm, $l_{1}=1.1\;\mathrm{mm}$, a=0.8mm, and $\beta =171.5^{\circ }$.

## 3. Kinematic Analysis of the Cable-Driven Continuum Robot

The kinematic analysis issues of the CDCR include the forward kinematics analysis, the differential kinematics analysis, and the inverse kinematics analysis. Since the CDCR can be considered as multiple CDCJMs connected in series, the kinematic analysis of the CDCR can be derived from the kinematic analysis of the CDCJM.

### 3.1. Kinematic Analysis of the Cable-Driven Continuum Joint Module

Two joint variables are introduced to simplify the kinematic analysis of the CDCJM, as shown in Figure 4. Based on the unique stiffness properties of low bending stiffness but high tensile and torsion stiffness, the bending shape of the optimized pattern-based flexible backbone can be considered as an arc in space with constant curvature. Therefore, the 2-DOF bending motions of the CDCJM can be expressed by two joint variables, including the bending direction angle $\alpha \in \left[0,2\pi \right]$ and the bending angle $\theta \in \left[0,\pi /4\right]$.

Two coordinate systems, $\left\{B\right\}$ and $\left\{E\right\}$, are defined for the kinematic analysis of the CDCJM. The origins and axis directions of $\left\{B\right\}$ and $\left\{E\right\}$ are depicted in Figure 5. The attachment points of the $i^{\mathrm{th}}$ cable at the base and moving platforms are denoted by $B_{i}$ and $E_{i}$, respectively. The plane $O_{b}AO_{e}$ is the bending plane of the CDCJM. $O_{b}B$ and $O_{e}E$ are the intersecting lines of the bending plane with the base and moving platforms, respectively.

#### 3.1.1. Displacement Analysis

The displacement analysis of the CDCJM is to derive the kinematic relationship between cable lengths and the pose of the moving platform. The relationship between driving cable lengths and the joint variables is computed by the closed-loop vector method. As shown in Figure 5, the $i^{\mathrm{th}}$ cable length is computed by the norm of the vector $\vec{B_{i}E_{i}}$:(1)$$\vec{B_{i}E_{i}}=\vec{B_{i}O_{b}}+\vec{O_{b}O_{e}}+\vec{O_{e}E_{i}}$$
where $\vec{B_{i}O_{b}}$, $\vec{O_{b}O_{e}}$, and $\vec{O_{e}E_{i}}$ are relative to the structure parameters and joint variables of the CDCJM. Their specific expressions can refer to [28].

The analytical expression of the $i^{\mathrm{th}}$ cable length is given by
(2)$$∥\vec{B_{i}E_{i}}{∥}^{2}={\left(r_{b}-r_{m}\right)}^{2}+4\sin^{2}\frac{\theta }{2}\left(\frac{L}{\theta }-r_{b}\cos\beta_{i}\right)\left(\frac{L}{\theta }-r_{m}\cos\beta_{i}\right)$$
where *L* is the length of the flexible backbone. $\beta_{i}=\alpha +\left(i-1\right)\pi /2$ is the rotation angle from $\vec{O_{b}B_{i}}$ to $\vec{O_{b}B}$. $r_{b}$ and $r_{m}$ denote the attachment points of cables fixed on the base and moving platforms, respectively.

According to (Equation 2), the expressions of two joint variables are computed by
(3)$$\begin{array}{cc} \alpha & =\arctan\frac{l_{2}^{2}-l_{4}^{2}}{l_{3}^{2}-l_{1}^{2}} \end{array}$$
(4)$$\begin{array}{cc} \frac{\left(1-\cos\theta \right)}{\theta } & =\frac{\sqrt{{\left(l_{3}^{2}-l_{1}^{2}\right)}^{2}+{\left(l_{2}^{2}-l_{4}^{2}\right)}^{2}}}{4L\left(r_{b}+r_{m}\right)} \end{array}$$

Referring to [28], θ is given as
(5)$$\theta =3.1136-\sqrt{9.6557-11.3636a}$$
where $a=\sqrt{{\left(l_{3}^{2}-l_{1}^{2}\right)}^{2}+{\left(l_{2}^{2}-l_{4}^{2}\right)}^{2}}/\left(4L\left(r_{b}+r_{m}\right)\right)$.

Based on the screw theory, the 2-DOF bending motions of the CDCJM can be described by the rotational movement around an instantaneous screw axis ξ. The rotational angle is θ. The pose of the moving platform is derived by the two-variable local Product-Of-Exponential (POE) formula
(6)$$T_{B,E}\left(\alpha ,\theta \right)=e^{\hat{\xi }\theta }T_{B,E}\left(0\right)$$
where $\hat{\xi }=\left[\begin{array}{cc} \hat{\omega } & v\\ 0 & 0 \end{array}\right]\in se(3)$ is the twist of the CDCJM. ω and *v* represent the directional vector and the position vector of ξ with respect to frame *B*, respectively.

In (Equation 6), $T_{B,E}\left(0\right)$ is the initial pose of CDCJM:(7)$$T_{B,E}\left(0\right)=\left[\begin{array}{cccc} 1 & 0 & 0 & 0\\ 0 & 1 & 0 & 0\\ 0 & 0 & 1 & L\\ 0 & 0 & 0 & 1 \end{array}\right]$$

Referring to [28], *v* and ω are given by
(8)$$v=L{\left[\begin{array}{ccc} -\frac{1}{2}\cos\alpha & -\frac{1}{2}\sin\alpha & \frac{1}{\theta }-\frac{1}{2}\cot\frac{1}{\theta } \end{array}\right]}^{T}$$
(9)$$\omega ={\left[\begin{array}{ccc} -\sin\alpha & \cos\alpha & 0 \end{array}\right]}^{T}$$

According to (Equation 8) and (Equation 9), the screw axis is uniquely determined by the joint variables.

#### 3.1.2. Velocity Analysis

As shown in Figure 4, the joint velocity is utilized for the velocity analysis of the CDCJM, in which the major issue is to compute the Jacobian matrix. The Jacobian matrix between the cable velocity and the joint velocity can be directly calculated according to (Equation 2), (Equation 3) and (Equation 5):(10)$$J_{l\Phi }={\left[\begin{array}{cccc} \frac{\partial l_{1}}{\partial \alpha } & \frac{\partial l_{2}}{\partial \alpha } & \frac{\partial l_{3}}{\partial \alpha } & \frac{\partial l_{4}}{\partial \alpha }\;\\ \frac{\partial l_{1}}{\partial \theta } & \frac{\partial l_{2}}{\partial \theta } & \frac{\partial l_{3}}{\partial \theta } & \frac{\partial l_{4}}{\partial \theta } \end{array}\right]}^{T}$$

Based on the Jacobian matrix $J_{l\Phi }$, the cable velocity can be derived as
(11)$$dl=J_{l\Phi }\;d\Phi$$
where $dl={\left[\begin{array}{cccc} dl_{1} & dl_{2} & dl_{3} & dl_{4} \end{array}\right]}^{T}$ is the cable velocity and $d\Phi ={\left[\begin{array}{cc} d\alpha & d\theta \end{array}\right]}^{T}$ is the joint velocity.

Based on [28], the instantaneous spatial velocity of the moving platform can be given by
(12)$${\hat{V}}_{B,E}^{B}={\dot{T}}_{B,E}\left(\alpha ,\theta \right)T_{B,E}^{-1}\left(\alpha ,\theta \right)$$
where ${\hat{V}}_{B,E}^{B}\in se(3)$ is a twist described in frame *B*, whose coordinates are formulated by $V_{B,E}^{B}=\left(v_{m}^{B},\omega_{m}^{B}\right)\in {ℜ}^{6\times 1}$. $\omega_{m}^{B}$ and $v_{m}^{B}$ represent the instantaneous angular velocity and the linear velocity of the CDCJM, respectively. Their specific expressions are given in [28]. ${\dot{T}}_{B,E}\left(\alpha ,\theta \right)$ is the derivative of $T_{B,E}\left(\alpha ,\theta \right)$ with respect to the joint variables.

According to (Equation 12), the linear velocity of point $O_{e}$ is derived by
(13)$$v_{O_{e}}^{B}=\omega_{m}^{B}\times p+v_{m}^{B}$$
where $p=\vec{O_{b}O_{e}}$.

Then, $J_{x\Phi }$ is formulated as
(14)$$J_{x\Phi }=\left[\begin{array}{cc} \frac{L\sin\alpha \left(\cos\theta -1\right)}{\theta } & \frac{L\cos\alpha \left(\theta \sin\theta +\cos\theta -1\right)}{\theta^{2}}\;\\ \frac{L\cos\alpha \left(1-\cos\theta \right)}{\theta } & \frac{L\sin\alpha \left(\theta \sin\theta +\cos\theta -1\right)}{\theta^{2}}\;\\ 0 & \frac{L\left(-\sin\theta +\theta \cos\theta \right)}{\theta^{2}}\;\\ -\cos\alpha \sin\theta & -\sin\alpha \\ -\sin\alpha \sin\theta & \cos\alpha \\ 1-\cos\theta & 0 \end{array}\right]$$

Therefore, the velocity of point $O_{e}$ can be calculated as
(15)$$dx=J_{x\Phi }\;d\Phi$$
where $dx={\left[v_{O_{e}}^{B},\omega_{m}^{B}\right]}^{T}$.

### 3.2. Kinematic Analysis of the Cable-Driven Continuum Robot

#### 3.2.1. Forward Kinematic Analysis

Due to the modular design approach, the forward kinematic model of the CDCR is derived from the product of the forward kinematic models of the CDCJMs:(16)$$T_{0,n}\left(\alpha ,\theta \right)=T_{0,1}\left(\alpha_{1},\theta_{1}\right)\cdots T_{i-1,i}\left(\alpha_{i},\theta_{i}\right)\cdots T_{n-1,n}\left(\alpha_{n},\theta_{n}\right)$$

According to (Equation 16), the pose of CDCR with respect to the base frame can be derived when given the number of the CDCJMs and their joint variables.

#### 3.2.2. Differential Kinematic Analysis

Based on (Equation 16), the instantaneous spatial velocity of the CDCR is given by
(17)$${\hat{V}}^{s}={\dot{T}}_{0,n}T_{0,n}^{-1}={\dot{T}}_{0,1}T_{0,1}^{-1}+T_{0,1}\left({\dot{T}}_{1,2}T_{1,2}^{-1}\right)T_{0,1}^{-1}+\cdots +T_{0,n-1}\left({\dot{T}}_{n-1,n}T_{n-1,n}^{-1}\right)T_{0,n-1}^{-1}$$

Substituting (Equation 12) into (Equation 17) and introducing the operator ∨, (Equation 17) can be rewritten as
(18)$$V^{s}=V_{0,1}^{1}+Ad_{T_{0,1}}V_{1,2}^{2}+\cdots +Ad_{T_{0,n-1}}V_{n-1,n}^{n}$$
where the operator ∨ denotes the mapping from $se\left(3\right)$ to ${ℜ}^{6\times 1}$. $V^{s}=\left(v^{s},\omega^{s}\right)\in {ℜ}^{6\times 1}$ is the twist coordinate of ${\hat{V}}^{s}$, in which $\omega^{s}$ and $v^{s}$ represent the spatial angular velocity and the spatial linear velocity of the CDCR with respect to its base frame, respectively.

$Ad_{T_{0,i}}\in {ℜ}^{6\times 6}$ is the adjoint transformation of $T_{0,i}$:(19)$$Ad_{T_{0,i}}=\left[\begin{array}{cc} R_{0,i} & p_{0,i}R_{0,i}\;\\ 0_{3\times 3} & R_{0,i} \end{array}\right]$$

Therefore, (Equation 18) can be represented by
(20)$$V^{s}=\left[\begin{array}{cccc} J_{1} & Ad_{T_{0,1}}J_{2} & \cdots & Ad_{T_{0,n-1}}J_{n} \end{array}\right]\dot{\phi }=J^{s}\dot{\phi }$$
where $J_{n}$ is the spatial Jacobian matrix of the $i^{\mathrm{th}}$ CDCJM. $\dot{\phi }={\left[{\dot{\phi }}_{1}^{T},\cdots ,{\dot{\phi }}_{n}^{T}\right]}^{T}$ is the joint velocity of the CDCR.

#### 3.2.3. Inverse Kinematic Analysis

The inverse kinematic analysis of the CDCR is to calculate the driving cable lengths when given the tip pose of the CDCR, which is significant for the trajectory planning of the CDCR. Referring to Figure 4, the inverse kinematic analysis of the CDCR is divided into two steps. The driving cable lengths can be calculated through (Equation 2) when given the joint variables of the CDCR. The analytical expression of the joint variables relative to the tip pose of the CDCR is difficult to derive since the CDCR is a hyper-redundant robot.

In this paper, the Newton-Rapson iteration method is employed. Given the desired pose $T_{0,n}^{d}$ of the CDCR and the initial guess of joint variables $\phi_{0}={\left[\alpha^{0},\theta^{0}\right]}^{T}$, the pose $T_{0,n}^{0}$ of the CDCR based on the initial guess of joint variables is derived through (Equation 16). The derivation between $T_{0,n}^{0}$ and $T_{0,n}^{d}$ is given by
(21)$$dx={\left(\log T_{0,n}^{d}{\left(T_{0,n}^{0}\right)}^{-1}\right)}^{\vee }$$

Based on the differential kinematics analysis, the differential changes in joint variables are computed by
(22)$$d\phi ={\left(J^{s}\right)}^{+}dx$$

Then, the joint variables are updated as
(23)$$\phi^{i+1}=\phi^{i}+d\phi^{i+1}$$
where the right superscripts represent the iterations.

According to (Equation 21)–(Equation 23), the joint variables are constantly updated until the error between the desired pose and the tip pose computed by the updated joint variables is within the allowable range.

## 4. Shape Sensing of the Cable-Driven Continuum Robot

### 4.1. Stretchable Capacitive Sensor

To achieve accurate motion control for the CDCR, a closed-loop motion control scheme is employed for each CDCJM, in which sensor feedback for the bending deflection of the flexible backbone, i.e., the bending angle and the bending directional angle, is essential. Since the flexible backbone of the CDCJM can achieve a large bending angle of $45^{\circ }$ with unlimited bending direction angles, the sensor has to possess the properties of a high stretch rate and high sensitivity in order to accurately detect the two joint angles. Among various strain sensors, the stretchable capacitive sensors made of flexible fabric and conductive nanomaterial are a class of elastic strain sensors with high sensitivity and high linearity for strain detection. As such, the stretchable capacitive sensor of Model RH-ESSA-01 from ElasTech is employed in this work, which has a maximum stretch rate of 50% and a minimum resolution of 0.05%.

The deflection of the stretchable capacitive sensor is the stretched deflection along its longitude direction. With the stretched deflection, the capacitance of the sensor will change. The employed stretchable sensors shown in Figure 6 have a length of 50mm and a width of 20mm, in which the effective sensing length is 40mm. The relationship between the stretched length of the sensor and its corresponding capacitance is given in Figure 7. The result verifies the employed stretchable sensor has high linearity. The analytical expression through the curve fitting is given by
(24)$$C=2.082\Delta l_{s}+61.75$$

### 4.2. Shape-Sensing Model of the Cable-Driven Continuum Joint Module

The shape-sensing model of the CDCJM is to derive the analytical expressions between the stretched lengths of stretchable sensors and the joint variables α and θ of the CDCJM. In this paper, a generic shape-sensing modeling method is proposed for the CDCJM, in which two joint variables can be measured through the stretched lengths of two stretchable capacitive sensors. Two stretchable capacitive sensors are located in arbitrary positions on the surface of the flexible backbone, and the location of each sensor is defined by its two endpoints. The width of the sensors is ignored in this paper, and each sensor can be simplified as a straight line. Since the surface of the flexible backbone without bending is a cylinder, the sensor can be considered as a helix along the cylinder, as shown in Figure 8. When the CDCJM produces the bending motions, the shape of the flexible backbone becomes a part of the torus, as shown in Figure 9. The location of each sensor varies from the bending deflections of the flexible backbone. The stretched length of a sensor is used to calculate the variation in the distance between two endpoints of the sensor on the surface of the backbone.

As shown in Figure 8, the two endpoints of the sensor are denoted by $S_{1}$ and $S_{2}$, respectively. An arbitrary point on the sensor is denoted by point *S*. Frame $\left\{B\right\}$ referring to Figure 5 is employed as the base frame of the flexible backbone. The intersection point between the $x_{b}$ axis of frame $\left\{B\right\}$ and the backbone surface is denoted by point A. The flexible backbone is unfolded into a plane $ABB^{\prime }A^{\prime }$ along the straight line AB. A rectangular coordinate system is established at point A. The helix in the plane $ABB^{\prime }A^{\prime }$ is a straight line $S_{1}S_{2}$, in which the angle between the x-axis and the line $S_{1}S_{2}$ is denoted by λ and the intercept of the line $S_{1}S_{2}$ on the y-axis is denoted by *b*. The helix is projected onto the bottom of the flexible backbone and the projected arc is denoted by $\overset{⌢}{S_{1}^{\prime }S_{2}^{\prime }}$. The projection point of point *S* is denoted by $S^{\prime }$. $t\in \left[t_{0},t_{0}+\Delta t\right]$ is the angle between the vector $\vec{O_{b}S^{\prime }}$ and the $x_{b}-$axis, in which $t_{0}$ is the angle between the vector $\vec{O_{b}S_{1}^{\prime }}$ and the $x_{b}-$axis. Therefore, the location of the sensor is defined by four parameters, i.e., *b*, λ, $t_{0}$, and Δt. *b* and $t_{0}$ are employed to determine the endpoint position of the sensor. λ is employed to describe the attachment angle of the sensor. Δt is employed to represent the length of the sensor. $a=\left[b,\lambda ,t_{0},\Delta t\right]$ is produced to denote the location of the sensor.

As shown in Figure 9, the intersection between the plane parallel to the bottom of the flexible backbone at point *S* and the neutral axis of the flexible backbone is denoted by point *F*. A local coordinate system $\left\{F\right\}$ is established at point *F*, in which the axes of frame $\left\{F\right\}$ are consistent with those of frame $\left\{B\right\}$ when the flexible backbone is in the initial state. The coordinates of the point *S* with respect to frame $\left\{F\right\}$ and frame $\left\{B\right\}$ are given by
(25)$$P_{S}^{F}\left(0\right)={\left[r\cos\left(t\right),r\sin\left(t\right),0\right]}^{T}$$
(26)$$P_{S}^{B}\left(0\right)=\left[\begin{array}{c} x_{S}^{B}\left(0\right)\\ y_{S}^{B}\left(0\right)\\ z_{S}^{B}\left(0\right) \end{array}\right]=\left[\begin{array}{c} r\cos t\\ r\sin t\\ r\cdot t\tan\lambda +b \end{array}\right]$$
where $r=\left(d_{1}+2t\right)/2$ is the radius of the flexible backbone.

When the flexible backbone produces the bending deflection, a coordinate system $\left\{N\right\}$ is introduced as a local frame, in which the origin is located at point *F*. The $N_{1}$ axis points to the center of curvature *M* and the $N_{3}$ axis is tangent to the neutral axis of the flexible backbone. The coordinates of point *S* expressed in frame $\left\{N\right\}$ becomes
(27)$$P_{S}^{N}\left(\alpha \right)={\left[r\cos(t-\alpha ),r\sin(t-\alpha ),0\right]}^{T}$$

The unit directional vectors of the $N_{1}$, $N_{2}$, and $N_{3}$ axes described in frame $\left\{B\right\}$ are derived by
(28)$$N_{1}=\left[\begin{array}{c} \cos\alpha \cos\left(\gamma \left(t\right)\right)\\ \sin\alpha \cos\left(\gamma \left(t\right)\right)\\ -\sin\left(\gamma \left(t\right)\right) \end{array}\right]$$
(29)$$N_{2}=\left[\begin{array}{c} -\sin\alpha \\ \cos\alpha \\ 0 \end{array}\right]$$
(30)$$N_{3}=N_{1}\times N_{2}$$

Therefore, the orientation matrix of frame $\left\{N\right\}$ is given by
(31)$$N=\left[N_{1},N_{2},N_{3}\right]$$

Based on the constant curvature assumption, the coordinate of point *F* described in frame $\left\{B\right\}$ is calculated by
(32)$$P_{F}^{B}\left(\alpha ,\theta \right)=\left[\begin{array}{c} x_{F}^{B}\\ y_{F}^{B}\\ z_{F}^{B} \end{array}\right]=\left[\begin{array}{c} R\left(1-\cos\left(\gamma \left(t\right)\right)\right)\cos\alpha \\ R\left(1-\cos\left(\gamma \left(t\right)\right)\right)\sin\alpha \\ R\sin\left(\gamma \left(t\right)\right) \end{array}\right]$$
where R=L/θ is the radius of the backbone bending curvature. $\gamma \left(t\right)=z_{0}\theta /L=\left(r\cdot t\tan\lambda +b\right)\theta /L$ is the angle between the line FM and the line $O_{b}M$.

According to (Equation 27), (Equation 32), and (Equation 31), the coordinates of the point *S* with respect to frame $\left\{B\right\}$ are derived by
(33)$$\begin{array}{cc} P_{S}^{B}\left(\alpha ,\theta \right) & =N\cdot P_{S}^{N}\left(\alpha \right)+P_{F}^{B}\left(\alpha ,\theta \right)\\ \; & =\left[\begin{array}{c} r\cos\left(-t+\alpha \right)\cos\left(\alpha \right)\cos\left(\gamma \left(t\right)\right)+r\sin\left(-t+\alpha \right)\sin\left(\alpha \right)+\\ R\left(1-\cos\left(\gamma \left(t\right)\right)\right)\cos\left(\alpha \right)\\ r\cos\left(-t+\alpha \right)\sin\left(\alpha \right)\cos\left(\gamma \left(t\right)\right)-r\sin\left(-t+\alpha \right)\cos\left(\alpha \right)+\\ R\left(1-\cos\left(\gamma \left(t\right)\right)\right)\sin\left(\alpha \right)\\ -r\cos\left(-t+\alpha \right)\sin\left(\gamma \left(t\right)\right)+R\sin\left(\gamma \left(t\right)\right) \end{array}\right] \end{array}$$

Based on the geometrical relationship shown in Figure 8, the initial length of the sensor is derived by
(34)$$l_{s}\left(0\right)=\frac{\Delta t\cdot r}{\cos\lambda }$$
where $l_{s}\left(0\right)$ represents the length of the sensor in the initial state.

Differentiating the both sides of (Equation 33), it becomes
(35)$$\begin{array}{cc} {\left(\frac{dP_{S}^{B}}{dt}\right)}^{2} & ={\left(\frac{\partial x}{\partial t}\right)}^{2}+{\left(\frac{\partial y}{\partial t}\right)}^{2}+{\left(\frac{\partial z}{\partial t}\right)}^{2}\\ \; & =\frac{{\left(\tan\lambda \right)}^{2}\theta^{2}r^{2}{\left(\frac{L}{\theta }-r\cos\left(t-\alpha \right)\right)}^{2}}{L^{2}}+r^{2}\\ \; & ={\left(\tan\lambda \right)}^{2}r^{2}{\left(1-\frac{r}{L}\theta \cos\left(t-\alpha \right)\right)}^{2}+r^{2}\\ \; & =r^{2}\cdot \left({\left(\tan\lambda \right)}^{2}{\left(1-k\theta \cos\left(t-\alpha \right)\right)}^{2}+1\right) \end{array}$$
where k=r/L.

Therefore, the stretched length of the sensor relative to α and θ is calculated by
(36)$$\begin{array}{cc} \Delta l_{s} & =\int_{t_{0}}^{t_{0}+\Delta t}dP_{S}^{B}dt-l_{s}\left(0\right)\\ \; & =r\int_{t_{0}}^{t_{0}+\Delta t}\left(\sqrt{{\left(\tan\lambda \right)}^{2}{\left(1-k\theta \cos\left(t-\alpha \right)\right)}^{2}+1}-\frac{1}{\cos\lambda }\right)dt \end{array}$$

According to (Equation 36), the numerical iteration method is employed to solve the joint variables when given the stretched lengths of sensors. Therefore, it needs to solve the Jacobian matrix between the stretched lengths of sensors and the joint variables. Differentiating both sides of (Equation 36), it becomes
(37)$$\frac{\partial \Delta l_{s}}{\partial \alpha }=r\left(g\left(\theta ,\alpha ,\lambda ,t_{0},k\right)-g\left(\theta ,\alpha ,\lambda ,t_{0}+\Delta t,k\right)\right)$$
(38)$$\frac{\partial \Delta l_{s}}{\partial \theta }=r\left(\int_{t_{0}}^{t_{0}+\Delta t}-\frac{\left(g{\left(\theta ,\alpha ,\lambda ,t,k\right)}^{2}-1\right)k\cos\left(t-\alpha \right)}{g\left(\theta ,\alpha ,\lambda ,t,k\right)}dt\right)$$
where $g\left(\theta ,\alpha ,\lambda ,t,k\right)=\sqrt{{\left(\tan\lambda \right)}^{2}{\left(1-k\theta \cos\left(t-\alpha \right)\right)}^{2}+1}$.

Assume that the number of stretchable capacitive sensors is *m*. The Jacobian matrix between the stretched lengths of sensors and the joint variables is defined as
(39)$$J_{a}=\left[\begin{array}{cc} \frac{\partial \Delta l_{s1}}{\partial \alpha } & \frac{\partial \Delta l_{s1}}{\partial \theta }\;\\ \frac{\partial \Delta l_{s2}}{\partial \alpha }\; & \frac{\partial \Delta l_{s2}}{\partial \theta }\;\\ \vdots & \vdots \;\\ \frac{\partial \Delta l_{sm}}{\partial \alpha }\; & \frac{\partial \Delta l_{sm}}{\partial \theta } \end{array}\right]$$
where $\Delta l_{si}\left(i=1,2,...,m\right)$ represents the tensile length of the $i^{\mathrm{th}}$ sensor.

When the flexible backbone produces the bending deflection, the stretched lengths of sensors $l_{s}^{a}={\left[l_{s1}^{a},\cdots ,l_{sm}^{a}\right]}^{T}$ are derived according to their capacitances. Given the initial guess of joint variables, the initial stretched lengths of sensors $l_{s}^{0}={\left[l_{s1}^{0},\cdots ,l_{sm}^{0}\right]}^{T}$ are calculated based on (Equation 36). Then, the derivation between $l_{s}^{0}$ and $l_{s}^{a}$ is calculated. The differential changes in joint variables are given by
(40)$$d\phi ={\left(J_{a}\right)}^{+}dl_{s}$$

The joint variables are updated referring to (Equation 23) until $d\Delta l_{s}$ is within the allowable range. In order to further verify the feasibility of the proposed shape-sensing model based on the numerical iteration method, a computation example is provided. In this example, two stretchable capacitive sensors are employed, such that $J_{a}\in {ℜ}^{2\times 2}$. The structure parameters of the flexible backbone and the locations of two sensors are given in Table 1.

The stretched lengths of the two sensors are given by $\Delta l_{s1}=-3.15\;\mathrm{mm}$ and $\Delta l_{s2}=-3.39\;\mathrm{mm}$. The initial guesses of the joint variables are α=0.76rad and θ=1.07rad. The average stretched length error of two sensors is calculated and its convergence result is shown in Figure 10. The updated joint variables are α=1.26rad and θ=1.57rad.

### 4.3. Calibration of Locations for Stretchable Capacitive Sensors

Due to the location errors of sensors, there exist derivations between the nominal stretched lengths and the actual stretched lengths, which will decrease the accuracy of the developed shape-sensing model. Therefore, it is necessary to calibrate the locations of sensors. Referring to (Equation 36), the stretched lengths of the sensors are determined by λ, $t_{0}$, and Δt. $d=\left[\lambda ,t_{0},\Delta t\right]$ is produced to denote the stretched parameters of the sensor that need to be calibrated. Based on (Equation 36), the derivatives of the stretched length for a sensor relative to λ, $t_{0}$, and Δt are calculated by
(41)$$\frac{\partial \Delta l_{s}}{\partial \lambda }=r\left(\int_{t_{0}}^{t_{0}+\Delta t}\left(\frac{\tan\lambda {\left(1-k\theta \cos\left(-t+\alpha \right)\right)}^{2}\left({\left(\tan\lambda \right)}^{2}+1\right)}{g\left(\theta ,\alpha ,\lambda ,t,k\right)}-\frac{\sin\lambda }{{\left(\cos\lambda \right)}^{2}}\right)dt\right)$$
(42)$$\frac{\partial \Delta l_{s}}{\partial t_{0}}=r\left(g\left(\theta ,\alpha ,\lambda ,t_{0}+\Delta t,k\right)-g\left(\theta ,\alpha ,\lambda ,t_{0},k\right)\right)$$
(43)$$\frac{\partial \Delta l_{s}}{\partial \Delta_{t}}=r\left(g\left(\theta ,\alpha ,\lambda ,t_{0}+\Delta t,k\right)-\frac{1}{\cos\lambda }\right)$$

The Jacobian matrix of the sensor stretched lengths relative to λ, $t_{0}$, and Δt is given by
(44)$$J_{b}=\left[\begin{array}{ccc} \frac{\partial \Delta l_{s1}}{\partial \lambda }\; & \frac{\partial \Delta l_{s1}}{\partial t_{0}}\; & \frac{\partial \Delta l_{s1}}{\partial \Delta t}\;\\ \frac{\partial \Delta l_{s2}}{\partial \lambda }\; & \frac{\partial \Delta l_{s2}}{\partial t_{0}}\; & \frac{\partial \Delta l_{s2}}{\partial \Delta t}\;\\ \vdots & \vdots & \vdots \;\\ \frac{\partial \Delta l_{sm}}{\partial \lambda }\; & \frac{\partial \Delta l_{sm}}{\partial t_{0}}\; & \frac{\partial \Delta l_{sm}}{\partial \Delta t} \end{array}\right]$$

Similarly, the numerical iteration method is utilized to derive the actual locations of sensors based on (Equation 44).

### 4.4. Closed-Loop Control of the Cable-Driven Continuum Robot

Based on the calibrated shape-sensing model, closed-loop control is conducted for the CDCR, as shown in Figure 11. In this paper, sliding mode control is employed as the control method for the CDCR, which is a robust control method that can effectively solve the control problem under parameter uncertainty.

For the CDCR, the dynamic equation expressed in the motor frame is given by
(45)$$M\left(q\right)\ddot{q}+C\left(q,\dot{q}\right)\dot{q}+G\left(q\right)=f\left(q,\dot{q}\right)+u$$
where $M\left(q\right)\ddot{q}$, $C\left(q,\dot{q}\right)\dot{q}$, and $G\left(q\right)$ are the generalized mass term, coriolis and centrifugal force terms, and the gravitational force term of the CDCR, respectively. $f\left(q,\dot{q}\right)$ is the frictional force term of the CDCR. u is the actuation force term of the motors. q is the rotation angle vector of the motors.

Given the desired pose $x_{d}$ of the CDCR, the desired rotation angle vector $q_{d}$ can be computed based on the inverse kinematic analysis. The rotation angle errors of the motors are denoted by $\tilde{q}=q_{d}-q$. Therefore, (Equation 45) can be rewritten as
(46)$$\begin{array}{cc} M\left(q\right)\ddot{\tilde{q}}+C\left(q,\dot{q}\right)\dot{\tilde{q}} & =G\left(q\right)-f\left(q,\dot{q}\right)-u+M\left(q\right){\ddot{q}}_{d}+C\left(q,\dot{q}\right){\dot{q}}_{d}\\ \; & =\eta -u \end{array}$$

The sliding mode surface is given as
(47)$$s=c\tilde{q}+\dot{\tilde{q}}$$
(48)$$\dot{s}=c\dot{\tilde{q}}+\ddot{\tilde{q}}$$
where $c\in {ℜ}^{m\times m}$ is a diagonal matrix. *m* is the number of motors.

Substituting (Equation 48) into (Equation 46), it becomes
(49)$$M\left(q\right)\dot{s}=M\left(q\right)c\dot{\tilde{q}}-C\left(q,\dot{q}\right)\dot{\tilde{q}}-u+\eta$$

Let $\mu =\eta -M\left(q\right)c\dot{\tilde{q}}+C\left(q,\dot{q}\right)c\tilde{q}+s$, (Equation 49) becomes
(50)$$M\left(q\right)\dot{s}=-s-u+\mu -C\left(q,\dot{q}\right)s$$

The Lyapunov function is constructed as
(51)$$V=\frac{1}{2}s^{T}M\left(q\right)s$$

According to (Equation 51), its differential equation is calculated as
(52)$$\dot{V}=s^{T}M\left(q\right)\dot{s}+\frac{1}{2}s^{T}\dot{M}s$$

Let $u=k\sup\left(s\right)+k_{p}s$, $\dot{V}$ is positive when k≥sup∥μ∥.

## 5. Experimental Results

### 5.1. Experimental Verification of the Cable-Driven Continuum Joint Module

The experiment of the CDCJM includes the stability measurement of the stretchable sensors and the accuracy test of the developed shape-sensing model under the proposed calibration method. The experimental prototype of the CDCJM is shown in Figure 12, which consists of motors, the prototype of the CDCJM, the motion capture system, and two stretchable sensors. The Qualisys motion capture system includes six cameras and markers, which have a measurement accuracy of 0.32mm. The type of camera is an Opus 500. The markers are evenly fixed on the base platform and the moving platform of the CDCJM, in which the coordinates of the markers are measured to calculate the pose of the CDCJM. The transformation matrix $T_{E}^{B}$ is calculated by
(53)$$T_{E}^{B}=T_{E}^{\mathrm{cam}}{\left(T_{B}^{\mathrm{cam}}\right)}^{-1}$$
where $T_{E}^{\mathrm{cam}}$ and $T_{B}^{\mathrm{cam}}$ represent the transformation matrices of frame $\left\{E\right\}$ frame $\left\{B\right\}$ with respect to the camera coordinate system, respectively.

In order to measure the stability of the stretchable capacitive sensors, the moving platform of the CDCJM follows a circle trajectory, in which the bending angle θ is a constant of 0.3rad and the bending direction angle α varies from 0rad to 2πrad. The nominal locations of two stretchable capacitive sensors are $d_{\mathrm{nom}1}=\left[4\pi /9,\pi /2,\pi /2\right]\;\mathrm{rad}$ and $d_{\mathrm{nom}2}=\left[-\pi /3,3\pi /4,\pi \right]\;\mathrm{rad}$, respectively. The initial length of each sensor is 50mm, and its prestretch length is 6mm. The capacitance changes in two stretchable sensors are shown in Figure 13. With the bending motions of the CDCJM, the capacitances of the two stretchable sensors vary with their stretched lengths. When the CDCJM returns to its initial pose, the capacitance changes in the two stretchable sensors are zero.

Furthermore, the calibration of the locations for two stretchable sensors is conducted to increase the accuracy of the developed shape-sensing model. When given the nominal joint variables, the nominal stretched lengths of two sensors and their corresponding capacitance changes can be calculated according to (Equation 36) and (Equation 24), respectively. Three experimental poses are shown in Figure 14. The actual capacitance changes are measured through the data collection module. The capacitance errors of two sensors between the nominal values and the measured values are shown in Figure 15 and Figure 16, which results from the location errors of the sensors. The average capacitance errors of the two sensors before calibration are 0.4433pF and 0.3916pF, respectively. Based on the numerical iteration method, the calibrated locations of two stretchable sensors are $d_{c1}=\left[1.3897,1.1887,1.5708\right]\;\mathrm{rad}$ and $d_{c2}=\left[-1.1545,2.9663,3.1416\right]\;\mathrm{rad}$, respectively. According to the calibrated parameters, the capacitance errors of the two sensors after calibration are reduced to 0.1123pF and 0.0679pF, respectively. Based on the calibrated shape-sensing model, the verification trajectory is introduced, in which the bending angle θ is a constant of 0.15rad and the bending direction angle α varies from 0rad to 2πrad. The capacitance errors of the two sensors between the calibrated values and the actual values after calibration are 0.0988pF and 0.0367pF, respectively. The errors of α and θ between the measured values and the actual values are about 0.03rad and 0.01rad, respectively.

### 5.2. Experimental Verification of the Cable-Driven Continuum Robot

In order to verify the effectiveness of the kinematics control method based on the calibrated shape-sensing model, the experiment of the CDCR is conducted. The prototype of the CDCR with multiple stretchable sensors is shown in Figure 17, which is composed of four CDCJMs. The flexible backbone of each CDCJM employs two stretchable sensors to measure its joint variables. The trajectory tracking accuracy of the CDCR is measured based on a circle trajectory, in which the radius of the circle is 55mm. As shown in Figure 18, the tracking errors of the CDCR under the open-loop control and the closed-loop control are 49.23 and 8.40mm, respectively. Compared with the tracking error under the open-loop control, the tracking error under the closed-loop control is reduced by 82.94%.

## 6. Conclusions

This paper proposes a generic shape-sensing modeling method for a Cable-Driven Continuum Robot (CDCR) with a flexible backbone. For the Cable-Driven Continuum Joint Module (CDCJM) with 2-DOF bending motions, its joint variables can be measured through stretched lengths of two stretchable sensors. Combined with the calibration of the locations for the sensors, the accurate measurement of the joint variables can be realized. The proposed shape-sensing modeling method has the advantages of good stability, high resolution ratio, high accuracy, and good antidisturbance ability. Based on the calibrated shape-sensing model, the sliding-mode-based closed-control method is implemented for the CDCR. The accuracy of the calibrated shape-sensing model is measured during the experiments, in which the capacitance errors of the sensors between the calibrated values and the actual values are 0.0988pF and 0.0367pF, respectively. The corresponding errors of the two joint variables α and θ are about 0.03rad and 0.01rad, respectively. According to the experimental result, the tracking error of the CDCR is reduced from 49.23mm to 8.40mm under the closed-loop control, which verifies the effectiveness of the proposed kinematic control method based on the calibrated shape-sensing model.

## Figures and Tables

**Figure 1 sensors-24-03385-f001:**
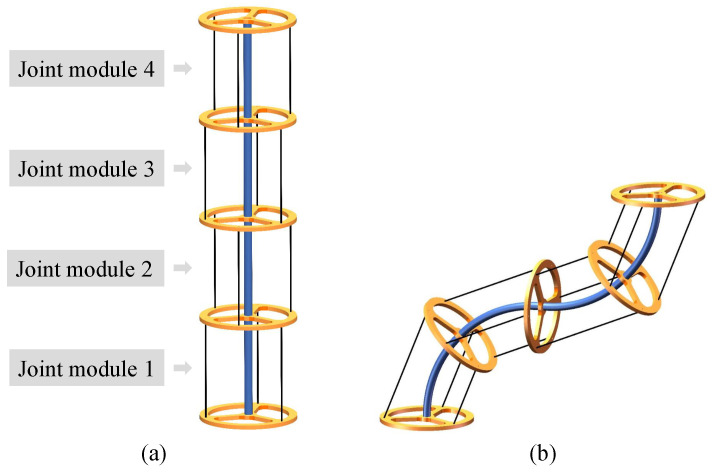
Schematic diagrams of a Cable-Driven Continuum Robot: (**a**) the initial pose without bending motions; (**b**) an arbitrary pose.

**Figure 2 sensors-24-03385-f002:**
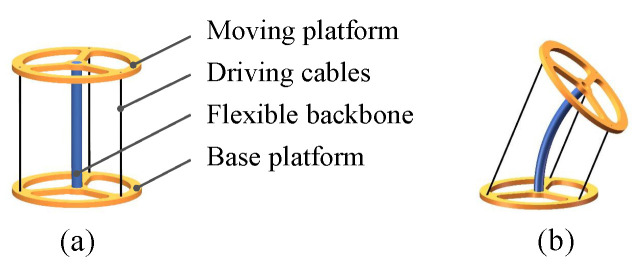
Schematic diagrams of a Cable-Driven Continuum Joint Module with three driving cables: (**a**) the initial pose without bending motions; (**b**) an arbitrary pose.

**Figure 3 sensors-24-03385-f003:**
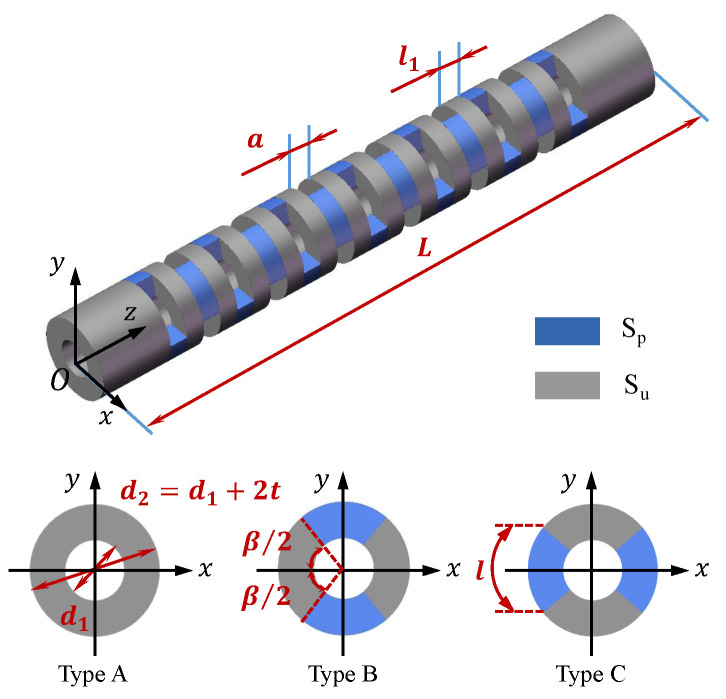
Schematic diagram and structure parameters of the pattern-based flexible backbone.

**Figure 4 sensors-24-03385-f004:**
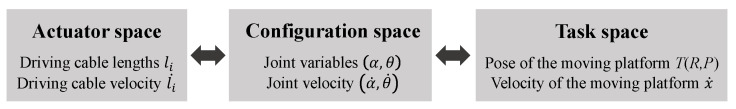
Three spaces and mapping of the Cable-Driven Continuum Joint Module.

**Figure 5 sensors-24-03385-f005:**
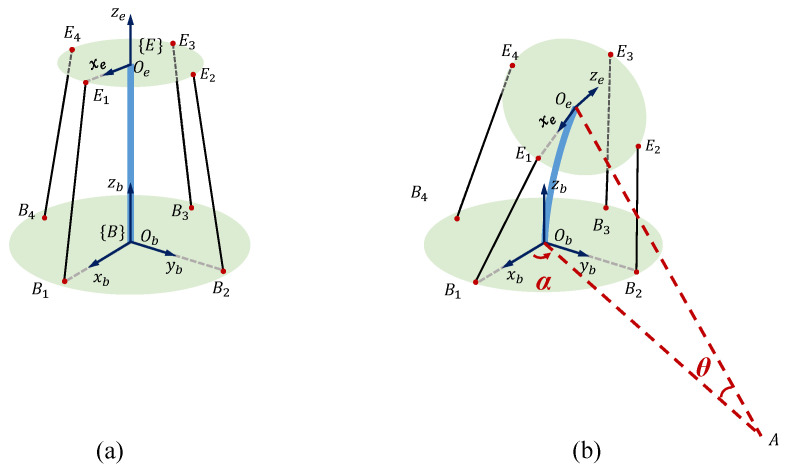
Kinematic diagrams of the Cable-Driven Continuum Joint Module: (**a**) the initial pose; (**b**) an arbitrary pose.

**Figure 6 sensors-24-03385-f006:**
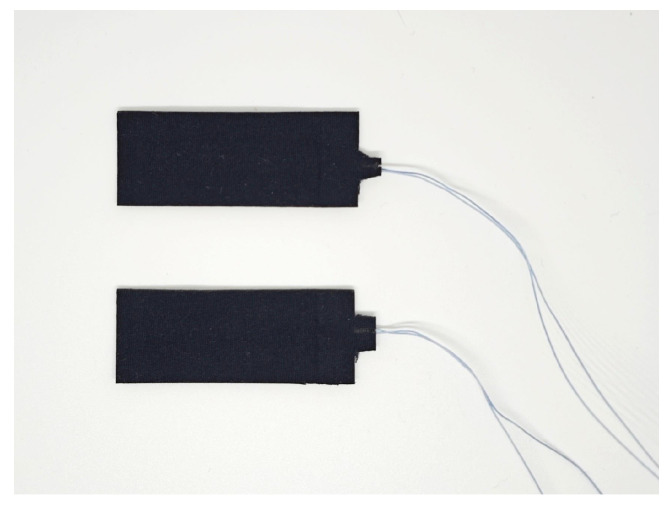
Prototypes of the stretchable capacitive sensors.

**Figure 7 sensors-24-03385-f007:**
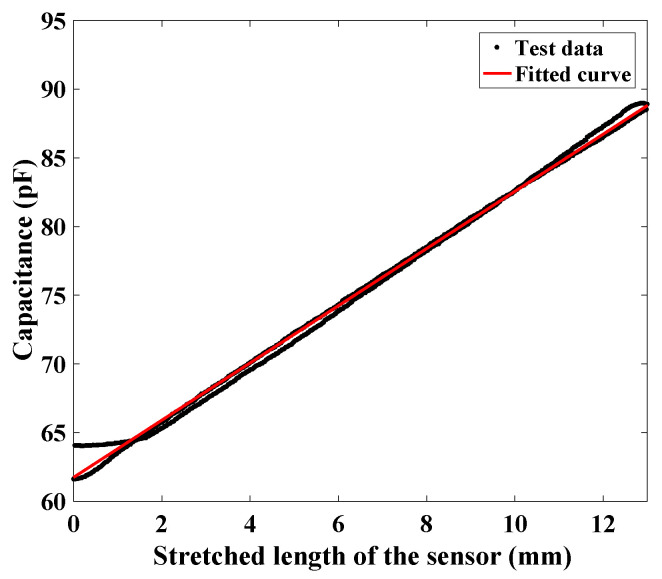
Relationship between the stretched length of a stretchable capacitive sensor and its corresponding capacitance.

**Figure 8 sensors-24-03385-f008:**
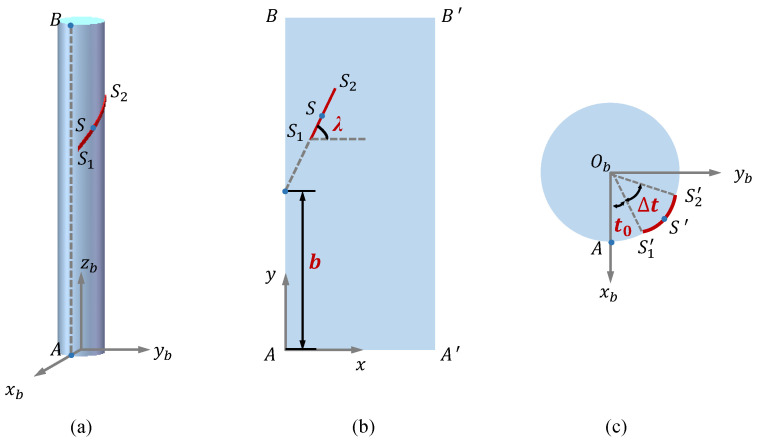
Schematic diagrams of the flexible backbone without bending and the corresponding projection views: (**a**) model of the flexible backbone attached with a stretchable capacitive sensor; (**b**) two-dimensional unfolded surface of the flexible backbone; (**c**) bottom plane of the flexible backbone and the corresponding projection of the sensor.

**Figure 9 sensors-24-03385-f009:**
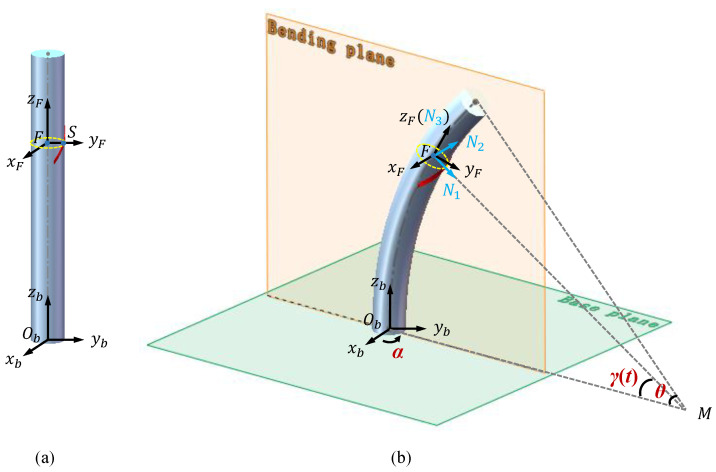
Schematic diagrams of the flexible backbone with a stretchable capacitive sensor in its bending state: (**a**) model of the flexible backbone without bending; (**b**) model of the flexible backbone with the bending deflection.

**Figure 10 sensors-24-03385-f010:**
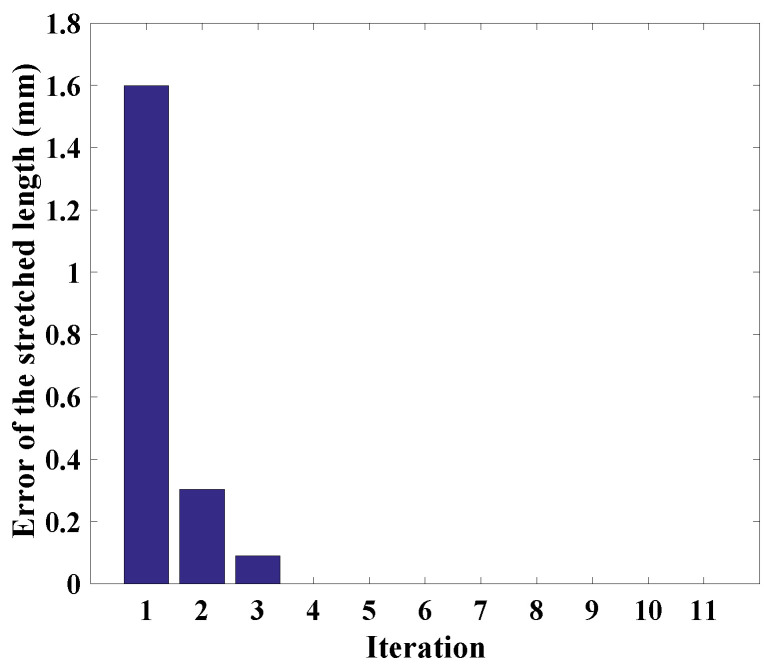
Convergence result of the stretched length error.

**Figure 11 sensors-24-03385-f011:**
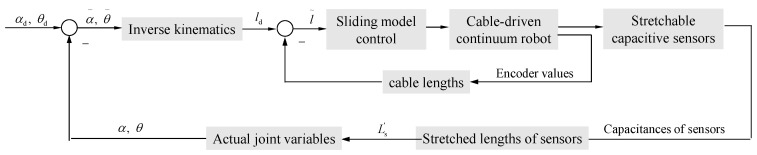
Block diagram of the closed-loop control method for the Cable-Driven Continuum Robot.

**Figure 12 sensors-24-03385-f012:**
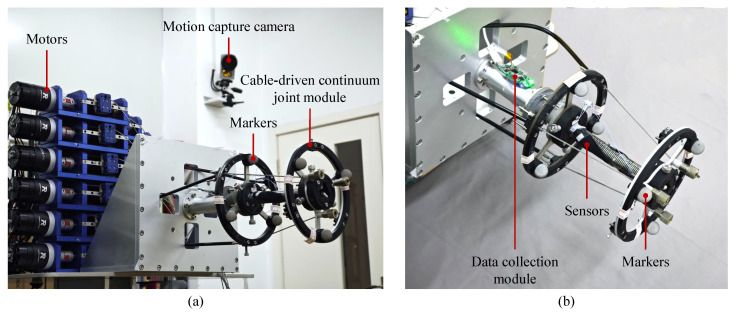
Experimental setup of the Cable-Driven Continuum Joint Module: (**a**) pose measurement system with motion capture cameras; (**b**) prototype of the Cable-Driven Continuum Joint Module with two stretchable capacitive sensors.

**Figure 13 sensors-24-03385-f013:**
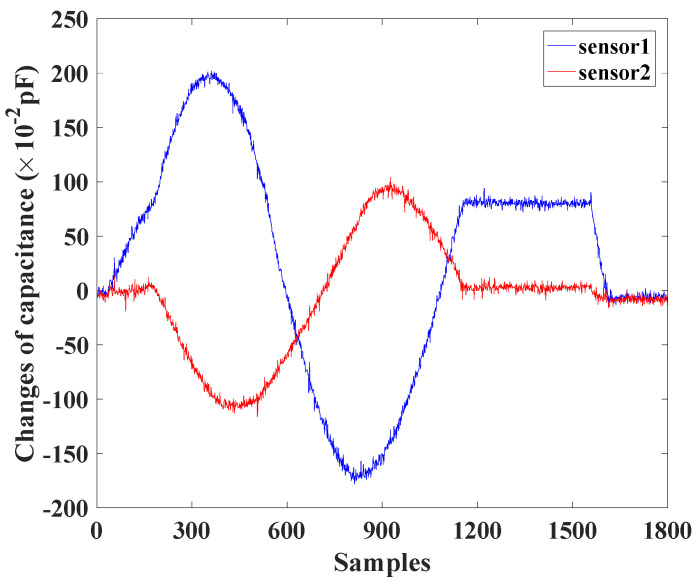
Capacitance of two stretchable sensors during the sensor stability test experiment.

**Figure 14 sensors-24-03385-f014:**
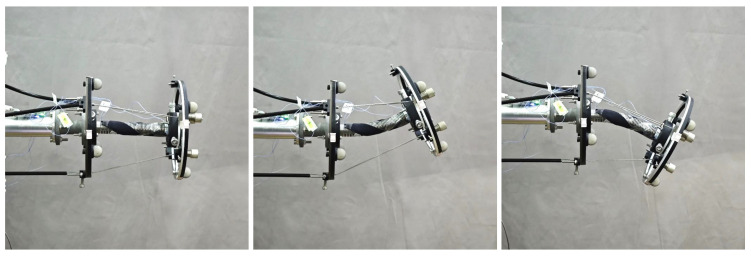
Experimental poses of the Cable-Driven Continuum Joint Module.

**Figure 15 sensors-24-03385-f015:**
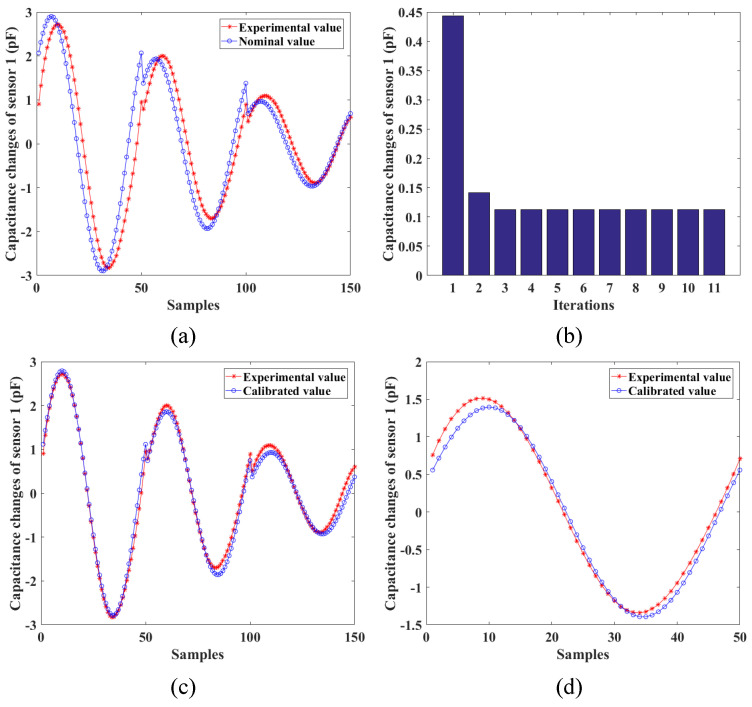
Calibration results of the first stretchable sensor: (**a**) capacitance errors before calibration; (**b**) convergence result of the capacitance errors; (**c**) capacitance errors after calibration; (**d**) capacitance errors of the verification trajectory after calibration.

**Figure 16 sensors-24-03385-f016:**
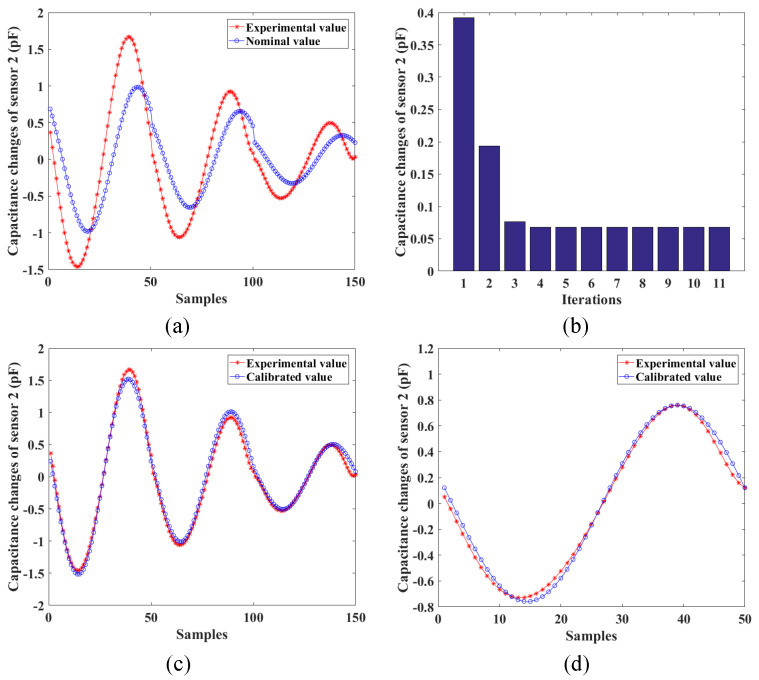
Calibration results of the second stretchable sensor: (**a**) capacitance errors before calibration; (**b**) convergence result of the capacitance errors; (**c**) capacitance errors after calibration; (**d**) capacitance errors of the verification trajectory after calibration.

**Figure 17 sensors-24-03385-f017:**
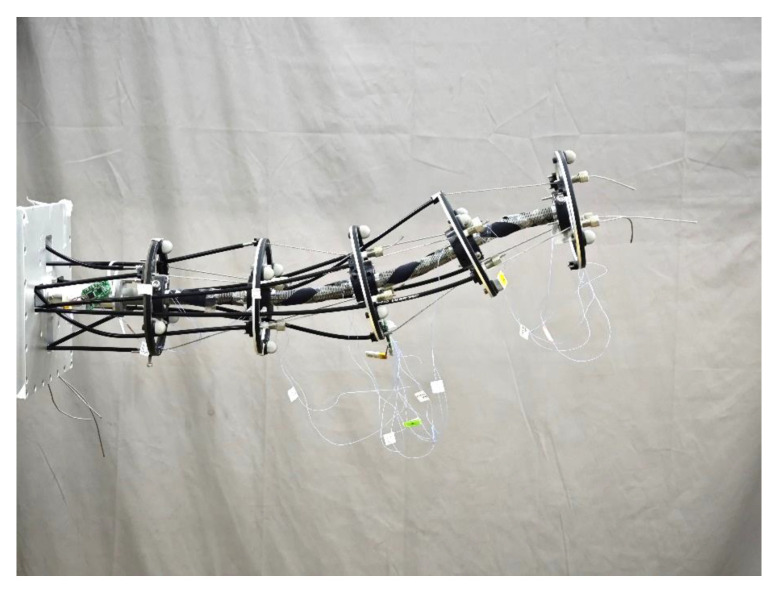
Experimental pose of the Cable-Driven Continuum Robot.

**Figure 18 sensors-24-03385-f018:**
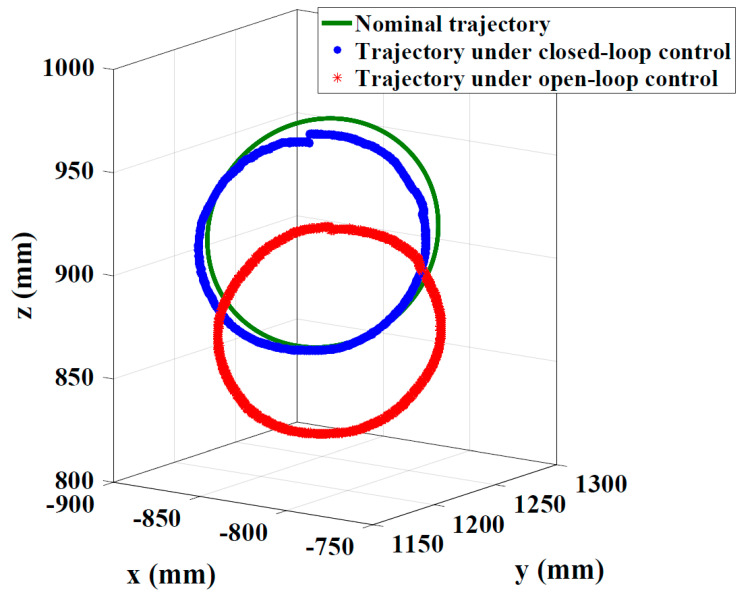
Trajectory tracking accuracy of the Cable-Driven Continuum Robot under open-loop control and closed-loop control.

**Table 1 sensors-24-03385-t001:** Structure parameters of the flexible backbone and the locations of the sensors for the computation example.

Property	Value
Flexible backbone length *L*	120mm
Flexible backbone radius *r*	10mm
Attachment position of the first sensor $a_{1}$	$\left[20\;\mathrm{mm},5\pi /12\;\mathrm{rad},\pi /2\;\mathrm{rad},\pi /3\;\mathrm{rad}\right]$
Attachment position of the second sensor $a_{2}$	$\left[50\;\mathrm{mm},2\pi /5\;\mathrm{rad},\pi /12\;\mathrm{rad},\pi /3\;\mathrm{rad}\right]$

## Data Availability

Data are contained within this article.

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
