# Peer review of "Shape Sensing and Kinematic Control of a Cable-Driven Continuum Robot Based on Stretchable Capacitive Sensors"

_sensors, 2024, doi:10.3390/s24113385_

Round 1

Reviewer 1 Report

Comments and Suggestions for Authors

Comments:

Clarity and Structure: The abstract's structure is clear, but when introducing CDCR and its constituent elements, more specificity can be provided, such as the features of CDCJM, the role of driving cables, etc. Terminology Explanation: Clear definitions or explanations should be provided when introducing new terms. For example, when mentioning "stretchable capacitive sensors," a brief explanation of their principles and functions can be given to ensure readers have a clear understanding of the paper's core concepts. Experimental Design and Results Presentation: The abstract mentions experimental results but lacks description of the experimental design and specific data. It is suggested to briefly describe the experimental design in the abstract and provide some quantified data of the main results to enhance the paper's readability and persuasiveness. Related Work: The abstract does not mention comparisons with existing literature. It is recommended to briefly introduce the relationship with existing technologies in the abstract to highlight the paper's innovation and contribution.

Revision Suggestions:

Provide more specific technical details: When introducing CDCR and its constituent elements, more specific technical details can be provided, such as the structural design of CDCJM, the working principle of driving cables, etc., to help readers better understand the technical contributions of the paper. Terminology Explanation: Provide concise explanations when introducing new terms to ensure readers have a clear understanding of the key terms in the paper. Increase description of experimental design and results: Add descriptions of the experimental design in the abstract, including experimental settings, equipment used, and methods employed. Additionally, provide some quantified data of the main results to support the conclusions in the abstract. Enhance comparison with related work: Add comparisons with related work in the abstract to highlight the paper's innovation and advantages.

Comments on the Quality of English Language

It is fine

Author Response

Thanls a lot for your significant comments and suggestions. We have carefully revised the manuscript. The specific responces are attached. We believe that the revised manuscript could meet the publication standard of the journal. Thank you very much!

Reviewer 2 Report

Comments and Suggestions for Authors

The authors of this paper present a Cable-Driven Continuum Robot (CDCR) that consists of a number of identical Cable Driven Continuum Joint Modules. In each CDCJM, a pattern based flexible backbone is employed as a passive compliant joint to generate 2-DOF bending deflections.

In this work, two stretchable capacitive sensors are employed to measure the bending shape of the flexible backbone. To further improve the accuracy of the shape sensing model, a calibration method is proposed to compensate the location errors of stretchable sensors.

The experimental tests demonstrate that with the calibration of the locations for the sensors, an accurate measurement of the joint variables is obtained. The tracking error of the cable-driven robot is reduced by almost 83%, so that it is verified the effectiveness of the proposed kinematic control method.

 Some comments:

-        The introduction is very complete and clear, emphasizing the advantages and weaknesses of the related methods in the literature.

-          The shape sensing of this type of flexible backbones is very significant for the cable-driven continuum robots, so this work constitutes a step forward in this field.

-          Line 348: correct “Jocabian”

-          Section 4.1: justify why the authors have selected this type of sensor.

-          The constant curvature assumption is applicable in practice? I mean, in reality does this also happen? Or is a theoretical assumption that implies a certain error in practice?

Author Response

Thanks a lot for your significant comments and suggestions. We have carefully revised the manuscript. The specific responces are attached. We believe that the revised manuscript could meet the publication standard of the journal. Thank you very much!
